# The Impact of Meteorological Parameters and Seasonal Changes on Reporting Patients with Selected Cardiovascular Diseases to Hospital Emergency Departments: A Pilot Study

**DOI:** 10.3390/ijerph20064838

**Published:** 2023-03-09

**Authors:** Paweł Kotecki, Barbara Więckowska, Barbara Stawińska-Witoszyńska

**Affiliations:** 1Department of Epidemiology and Hygiene, Chair of Social Medicine, Poznan University of Medical Sciences, 60-806 Poznań, Poland; 2Department of Computer Science and Statistics, Poznan University of Medical Sciences, 60-806 Poznań, Poland

**Keywords:** linear regression model, meteorological parameters, weather, seasonality of changes, cardiovascular diseases, emergency departments, Poznań

## Abstract

(1) Background: So far, research results have confirmed the relationship between heat and cold stress, the fluctuations in atmospheric pressure and high relative humidity, and the vulnerability of patients with so-called “weather-dependent” diseases which could lead to death. This study aimed to determine the meteorological parameters, their interactions, and the seasonal changes of the most significant factors in predicting the number of patients reporting to the Emergency Departments (EDs) in Poznań (Poland) during 2019. (2) Methods: The analysis included the meteorological parameters and data of 3606 patients diagnosed with essential or complicated arterial hypertension, myocardial infarction, chronic ischemic heart disease, and ischemic or unspecified stroke by the International Classification of Diseases (ICD-10). The meteorological data (days per week and seasonal data) were used to build a linear regression model to assess the changes in the daily number of reporting patients. The input data for the final model were selected based on the principal component analysis (PCA), and built for each delay and acceleration (reporting up to 3 days before the change or up to 3 days after the change of the meteorological parameter). (3) Results: A significantly lower number of reports was observed during weekends compared to working days (*standardised b* = −0.254, *p*-value < 0.0001) and three days before the maximum daily air temperature in the spring and summer period (*standardised b* = −0.748, *p*-value < 0.0001), while two days after the increase in the daily amplitude of atmospheric pressure (*standardised b* = 0.116, *p*-value = 0.0267), and also on the day of occurrence of the unfavourable interdiurnal air temperature change, an increase in the number of patients was noted (*standardised b* = 0.115, *p*-value = 0.0186). The changes in the last two parameters were statistically insignificant. Based on the obtained results, the negative impact of the changes in the meteorological conditions on the number of reports to the EDs in Poznań was determined.

## 1. Introduction

In recent years, the risk posed by climate change and its effects and human adaptation to climate fluctuations have been featured in social and scientific discussions. Climate change is assumed to be anthropogenic, and global warming, attributed mainly to the excessive emissions of greenhouse gases, has already affected all land, oceans, and the atmosphere. Due to the increase in the global average air and ocean temperatures, snow and ice have melted, and sea levels have started to rise. It is expected that climate changes, often visible in the form of extreme weather phenomena (e.g., heat waves, droughts, heavy rainfall, hurricanes, or tropical cyclones), that are occurring now more often than previously, will increase in frequency and intensity [1]. Climate change, usually unfavourable, poses environmental risks, and may harm socio-economic activity and human health. Estimating the well-being consequences of climate change is very difficult, because, apart from its direct effects, also notable are its secondary and tertiary effects [2]. Its immediate results are mainly related to changes in the frequency of extreme weather conditions (e.g., heat, drought, and floods, etc.). Its indirect effects are related to the environment (air pollution, water or food contamination, and changes in infectious disease patterns) or human activity, e.g., migrations, stress, and social conflicts [3,4]. The need to disseminate knowledge about the risks resulting from climate change, especially to the residents of areas at risk of extreme weather events, is the task of the representatives of many fields of science (e.g., medicine, climatology, biology, physics, chemistry, ecology, engineering, including information technology engineering, economics, or psychology) because climate change is an interdisciplinary phenomenon [5,6]. The research results confirm the relationship between heat and cold stress, atmospheric pressure fluctuations and high relative air humidity, and the severity of the disease symptoms in people suffering from so-called “weather-related” diseases that could even lead to death.

We decided to look at cardiovascular diseases because these diseases are the most common cause of death in Poland. The Polish population, compared to the European population (EU-28), is characterised by mortality that is almost 2 times higher for ischemic heart disease (Poland:131 deaths/100,000 inhabitants; and “EU-27”: 7 deaths/100,000 inhabitants) and 1.5 times higher for stroke (respectively, 53 deaths/100,000 compared to 38 deaths/100,000 inhabitants) [7]. The influence of seasonal changes, air temperature, or atmospheric pressure on the number of admissions of patients with symptoms of cardiovascular diseases to hospitals was the subject of many studies, which usually showed a relationship between the season and the days of the week, as well as air temperature and an increased incidence of cardiovascular diseases [8,9,10,11]. 

The study aimed to determine the meteorological parameters, their interactions, and the seasonal changes of the most significant factors in predicting the number of patients with selected cardiovascular diseases that reported to the EDs in Poznań in 2019 (Poland). As a result, a comprehensive assessment of the interaction between the season, the day, the changes in the values of the various meteorological parameters, and the actual and predicted average daily number of cases registered in the EDs’ systems was created. 

The results of this project are helpful to Polish public health officials, hospital authorities, and healthcare professionals to increase their knowledge about the impact of the weather on patients’ health.

## 2. Materials and Methods

Poznań is a city located in western Poland, and its climate is in the transition zone between a humid continental climate and a moderate oceanic climate [12].

In presented study, 2019 was regarded as the most specific year of research, and of a high impact for Poznań in terms of the meteorological parameters over the 2010–2019 decade. Therefore, in 2019, a large number of particular days, significant from bioclimatological point of view, were observed and defined as stimulus days, which were determined based on the daily minimum and maximum air temperature, and the most significant number of stimulus days were classified as days with a dangerous interdiurnal pressure change of atmospheric pressure over the 2010–2019 decade.

The meteorological data for all the days of 2019 were obtained from the Poznań–Ławica meteorological station [13], which is a part of the measurement network of the Institute of Meteorology and Water Management National Research Institute (IMiGW). The legal basis for sharing this data is contained in the Act of 25 February 2016 on the re-use of public sector information [14].

Table 1 presents the analysed thermal parameters, barometric parameters, and hygrometric parameters used in this study.

The collected medical database (see Table 2) consisted of 3606 patients with one of the following diagnoses: essential or complicated arterial hypertension, myocardial infarction, chronic ischemic heart disease, cerebral infarction, or ischemic or unspecified stroke.

The classification of the diseases was made following the current “International Statistical Classification of Diseases and Health Problems” (ICD-10) [15]. All the selected disease entities were assessed jointly as one group. The data cover only the daily number of reports of patients with the latter diagnosis registered in Emergency Departments (EDs) in Poznań, as obtained from the National Health Fund (NFZ) database. The medical data were obtained based on the daily reports sent by hospitals in Poznań, containing the number of diagnoses of individual disease entities. Each hospital sends statistical notifications to the National Health Fund (NFZ). The data were obtained only for scientific purposes and for the possibility of using the results to benefit public health [16].

### 2.1. Methods

#### 2.1.1. Principal Component Analysis (PCA)

Due to the large number of mutually correlated meteorological parameters, which could potentially affect the number of reports to medical facilities, a separate initial selection was carried out in the subgroup of thermal parameters and the subgroup of barometric parameters (see Table 1) using a principal component analysis (PCA). The validity of this method was confirmed with the Bartlett test [17], which assessed the significance of the mutual correlation of the meteorological parameters, and the Kaiser–Mayer–Olkin (KMO) coefficient was used to check the degree of this correlation. A scree plot was used to select the number of components (the parameters from both subgroups) used in the PCA. The eigenvalues were chosen to determine the number of elements and to inform about what percentage of the variability was explained by a given component. The first component described the most significant part of the variability, which decreased with each subsequent component. The scree plot presents the eigenvalues from the largest to the smallest, and its flattening from a specific value is taken as the cut-off point for the number of components. The components to the right of this point were omitted because they represented negligible variance and primarily random noise. For the eigenvalues to the left of the designated point, the factor loads were determined, reflecting the influence of the individual meteorological variables on a given principal component. Based on the factor loadings and the matrix of correlations of the meteorological variables, the thermal parameters and barometric parameters were selected, which were the basis for the regression models built in the subsequent stages.

The autocorrelation of the score and the dependence of the covariates on time cannot be adequately modelled with a simple linear regression model. Therefore, the authors are aware of some underestimation in their results and will look for statistical methods to increase the accuracy of their analyses. 

In building the linear regression model, short-term seasonality was taken into account, related to the days of the week (weekend included/weekend excluded) and long-term-associations with the warm season (spring and summer) and cold season (autumn and winter). To assess the impact of the weather on the patient directly, i.e., on the day when the specific weather conditions occurred, or with a delay/acceleration, seven linear regression models were built step-by-step (using a step-down method). The models aim to assess the impact of seasonality, the previously selected (using the PCA method) thermal and barometric parameters, and the hygrometric parameter on the total number of daily reports of the diseases chosen. The seven models include:The one model of the direct impact of the weather on the patient, the so-called “D0”.The three models of the so-called accelerations (“D-1”, “D-2”, and “D-3”), i.e., the models determining the impact of the weather changes on the patient up to three days before the day that may adversely affect their well-being and health.The three models of delays (“D + 1”, “D + 2”, and “D + 3”), i.e., the models determining the impact of the weather changes on the patient up to three days after the occurrence of a day that may adversely affect their well-being and health.

Based on the seven models that were reduced in steps, a ranking was created that allows for a selection of the number of delays/accelerations for the individual meteorological parameters that correlate most strongly with the number of hospital cases. The choice was based on comparing the beta coefficients (slope coefficients) obtained for the lags/accelerations of the individual meteorological variables and their interaction with the variables describing seasonality. The final regression model included proper accelerated/delayed meteorological variables that were selected in this way, along with information on short- and long-term seasonality and its significant interactions.

We resigned from using classic ARIMA models (autoregressive integrated moving average model) due to the analysis of both the time delays and accelerations. Instead, we used a dynamic regression model, i.e., considering the time delays and time accelerations, and included the autocorrelation assumption in the final regression model.

Before including the variables in the model, we examined their relationship with the dependent variable. The variables that showed a non-linear relationship, e.g., the day of the week and the season, were grouped into binary variables. The effect of both high and low temperatures on the number of reports was non-linear, as it was modified by the season. The impact of low temperatures was more significant in winter and high temperatures were more significant in summer, considering that the interaction of the season and temperature allowed for the assumption of linearity in the subgroups determined by these interactions. Contrary to our concerns, the relationship between air pressure and the number of cardiovascular disease reports was linear.

#### 2.1.2. The Final Regression Model

The final model was checked for compliance with the assumptions. Due to residual autocorrelation (the correlation of residuals for the first lag), it was extended with lagging residuals. Using the F-test, the final model was compared, taking into account seasonality, the accelerated/delayed meteorological variables, and the interactions with the minimal model, based only on seasonality, i.e., the day of the week and the time of the year, in terms of their quality of predicting the number of calls to medical facilities. The final model is described with its coefficients, the significance of the individual components, and the coefficient of determination evaluating the quality of the fit. The dependencies that were obtained in the model are presented in graphs, showing the averages with a 95% confidence interval for the categorical variables and the regression line with a 95% confidence interval for the continuous variables.

A level of statistical significance of 0.05 was assumed in all analyses. The calculations were made using the PQStat v1.8.4 program.

## 3. Results

### 3.1. Principal Component Analysis (PCA)

The percentage of variance explained by the individual eigenvalues is presented in the scree plot (Figure 1). The vertical line marks the end of the scree, which means that the number of two principal components explains the variance of more than 86% of the thermal parameters and more than 82% of the barometric parameters (Figure 1). The Bartlett test for both types of the meteorological parameters showed a statistically significant correlation of these variables (*p* < 0.0001), the strength of which is determined by the relatively mediocre and miserable values of the Kaiser–Mayer–Olkin coefficient (KMO = 0.66 for the thermal parameters and KMO = 0.59 for the barometric parameters).

The determined factor loadings of the individual thermal and pressure variables in the first two principal components are presented in Table 3. Finally, the mutual correlations of the thermal and pressure variables are shown in Table 4. In the subgroup of the humidity parameters, there was only one variable, i.e., the average daily relative air humidity, which was automatically used in the built regression models.

The maximum daily air temperature was represented most strongly in the first component. It was connected by a strong correlation (Pearson’s correlation coefficient > 0.7) with all the other thermal parameters, except for the inter-day change in air temperature. The inter-day change in the air temperature was most strongly represented in the second component (factor load value 0.97). For this reason, these two variables from the thermal factors were selected to be used in a compilation of the final regression model. In the case of the correlation matrix of the barometric parameters: the average daily atmospheric pressure, the maximum daily atmospheric pressure, and the minimum daily atmospheric pressure were highly correlated and strongly represented in the first component. Therefore, only one of these factors (the mean daily atmospheric pressure) represented the first component. The logarithm value of the daily atmospheric pressure amplitude was not strongly correlated with the other variables. Still, it was linked by a strong relationship with the second principal component and expressed the second component in the built regression models.

### 3.2. Stepwise Reduced Regression Models

Table 5 presents the value of the *b-coefficient,* and deceleration/acceleration coefficients with the highest values are marked. In this way, the variables selected for the final model were indicated.

As a result of the stepwise reduction, the parameters of the average daily atmospheric air pressure and the average daily relative air humidity were removed from the subsequent analyses.

### 3.3. The Final Regression Model

As a result of the use of stepwise regression models, the final regression model was created based on the following variables: weekend (no = 0/yes = 1), season (cold = 0/warm = 1), maximum daily air temperature (°C) with the acceleration of three days (D-3), interdiurnal air temperature change (°C) on the date of diagnosis (D0), logarithmic to the value of the daily amplitude of atmospheric pressure with a delay of two days (D + 2), *x2 * x3* = interaction (product) of parameters season (0/1), and the maximum daily air temperature (°C) with the acceleration of three days (D-3).
Number of diagnoses = *b*_0_ + *b*_1_*x*_1_ + *b*_2_*x*_2_ + *b*_3_*x*_3_ + *b*_4_*x*_4_ + *b*_5_*x*_5_ + *b*_6_*x*_6_ + *b*_7_*r*

The coefficients *b*0,…, *b*7 and the standardised equivalents of these coefficients, showing the impact of the selected parameters of the final model on the number of applications to the EDs, are presented in Table 6.

The most significant impact on the number of reports (the highest “*standardised b*” value) was observed for the season (cold/warm) and for the maximum daily air temperature with an acceleration of three days (Table 6).

The average daily number of the diagnoses of the analysed group of diseases, in total, was 10 (Table 2). To present the forecast quality of the final regression model, the results were compared with the observed number of daily notifications. The average daily number of reports decreased by almost 2 on weekends (an average decrease of 1.79 for the observed frequencies and 1.82 for the forecasted frequencies, Figure 2a), and slightly more than 1 report during the warm season (an average reduction of 1.42 for the observed and predicted frequencies data, Figure 2b). The interaction of the season and the maximum daily air temperature was also noted. This interaction indicated a decrease in the number of registered patients three days before the increase in the maximum daily air temperature in the warm season, and an increase in the winter season (Figure 2c), while the rate of increase/decrease in the number of diagnoses was almost the same for the observed and predicted numbers. Changing the daily air temperature on the date of diagnosis increased the number of patients reporting to EDs on the same day (Figure 2d). Similarly, an increased number of reports was expected two days after the increase in the amplitude of the atmospheric pressure. The rate of the changes in the number of diagnoses for the observed data and the data predicted by the model was very close, and the drawn regression lines coincided (Figure 2e).

The final model enriched with the weather parameters was significantly better than the minimal model containing only the seasonal information (weekend and season) (Table 7). For example, the changes in the number of patients were explained by the weather factors and seasonality in less than 17% of patients, while the model built solely on the seasonal factors explained these changes in less than 11% of patients.

The authors are aware of the low value of the R^2^ and, therefore, will look for solutions to improve the specification in subsequent works that will help the model explain more variability in the data. The model explains the change in the number of patients presenting to a small extent, it is true. However, based on the time parameters (the day of the week and season of the year) and the weather, it is difficult to predict the number of patients with a high precision, with a person falling ill and reporting back depending on many factors, e.g., the patient’s lifestyle or genetic burden, which we have not examined here.

However, the authors have shown that adding the weather variables (indicated in Table 6) to the model improves its fit and is statistically significant.

## 4. Discussion

In recent years, an increased number of scientific papers have been published on the adverse impacts of weather and climate on the human body. There were also publications with research results on the correlation between the variability of selected meteorological parameters and the number of patients with various CVD symptoms. An example of this may be the work of Rusticucci et al. [18], which describes the relationship between weather changes in the summer and winter and the number of people reported to the emergency room of a hospital in Buenos Aires. The study’s authors distinguished seven diagnostic groups. Each concerned patients with different diagnoses of disease entities, indicated between summer and winter. Group 1: respiratory, cardiovascular, and chest pain; Group 2: digestive, genitourinary, and abdominal complaints; Group 3: neurological and psychopathological disorders; Group 4: infections; Group 5: ailments associated with bruises and the crushing of bones and muscles; Group 6: skin and allergies; and Group 7: different complaints. The results showed, in general, a higher number of notifications in the winter period by approx. 16.7%, compared to the summer period, which could be related to the holiday period. In the summer, only the number of reports related to diseases of the digestive and genitourinary systems and abdominal pains, isolated separately, and people from group 6 with skin lesions and allergies, increased. The significant difference between extremely high temperatures and wet days or days with a high air humidity, and the deterioration of the health of the patients from group 1 (respiratory, cardiovascular, and chest pain), as described in the Argentinian studies, was consistent with the results of the studies presented by Lecha [19], who showed significant dependencies between the incidence of cardiovascular diseases and the so-called heat stress in Cuba. Although the average air temperature during the winter in Buenos Aires is higher than in Poznań, and is additionally accompanied by a 95% humidity, the results of both studies were consistent in terms of an increase in the number of people reporting to the hospital with symptoms of cardiovascular diseases in the winter, as well as a decline of the reports on weekends. The Argentinian study’s authors associated the latter with trips out of town or postponing reporting to the hospital until Monday. This suggestion may also fit with the results of the analysis of the present manuscript. The Norwegian authors Tollefsen and Dickstein [20] drew similar conclusions. They explained the lower number of calls on weekends and the high number on Mondays by the behaviour of some patients waiting for a doctor’s appointment until Monday morning, due to symptoms that developed over the weekend. It is also possible that the high number of admissions on Mondays was associated with lesser access to doctors on weekends. Still, according to the opinion of the Norwegian researchers, this fact should not affect the number of emergency calls. Several papers on sudden changes in meteorological parameters, their potential impact on the mortality rate in the Czech Republic, and the number of hospitalisations due to cardiovascular diseases in Prague, were published by Plavcová and Kyselý [21,22,23]. Their research showed a significant increase in the mortality rate following large increases in temperature and during days of large decreases in atmospheric pressure, and a decrease in mortality after large decreases in temperature, an increase in atmospheric pressure, and after the passage of strong atmospheric cold fronts. The same authors showed an increase in the number of patients in the emergency rooms of Prague hospitals during the winter on days with a significant drop in atmospheric pressure. Such changes were not observed in summer, nor with pressure increase, regardless of the season. Sudden changes in blood pressure were also associated with an increase in patients’ mortality level with cardiovascular diseases during winter. Despite the Czech Republic’s and Poland’s climatic similarity, the Poznań and Prague studies’ results differed. In Poznań, a significantly lower number of cases of calls to the EDs was observed three days before the increase in the maximum daily air temperature during the warm season. However, an increase in the number of cases was noted in the winter season, especially on the day of significant change in the daily air temperature. Similarly, an increased number of reports was expected two days after the increase in the amplitude of atmospheric pressure, while in Prague, this was in winter, on days with a significant decrease in atmospheric pressure. In Norway, all-cause mortality, the incidence of coronary heart disease, and the mortality from this cause increased with the decrease in temperature. The work already cited (Tollefsen and Dickstein [16]), based on the analysis of patient reports in the Norwegian administrative district of Rogaland, showed a significant relationship between the occurrence of days with snow or rain and an increased number of patient reports. The results of the Finnish study by Sohail et al. [24], conducted in the summer months of 2001–2017, suggested a link between heat waves and the occurrence of certain subtypes of cardiovascular diseases. A borderline significant relationship was confirmed between hot days and the increase in admissions due to cerebrovascular diseases in people of all age groups to hospitals in Helsinki. The authors of these studies noted that there was no evidence in the previous scientific literature of an effect of heat or heatwaves on the hospital admissions of people with these conditions. A review of cross-sectional studies also found no such association [25,26]. The positive association between heatwaves and myocardial infarction admissions found in the 65–74 age group is in contrast with previous studies that found no or a reduced risk of hospital admissions during the summer months [26,27,28]. A lower risk of hospital admissions due to myocardial infarction in the summer was proven by Swedish researchers working on the SWEDEHEARTH Project, who found that the following weather parameters were associated with an increased risk of myocardial infarction among Swedish residents: low air temperature, low-pressure atmospheric conditions, high wind speed, and a shorter duration of sunshine. The greatest correlation was observed in the case of changes in the air temperature [29]. An increase in the air temperature of one standard deviation (7.4 °C) was associated with a 2.8% reduction in myocardial infarction risk. The MI (the unadjusted incidence ratio) was 0.972; 95% CI, 0.967–0.977; *p* < 0.001. The results of retrospective studies conducted in Hungary were not identical to those in Sweden. Indeed, the lowest number of myocardial infarctions was recorded in the summer, but their peak incidence was in the spring, and these differences were significant [30]. In the scientific literature, studies have also analysed the impact of the delay associated with air masses generating weather changes on health. The results of studies carried out in hospitals in Florence during the winter period [31] showed a sudden increase in the admissions of patients with myocardial infarction 24 h after a day with high pressure, and six days after a day characterised by a low-pressure system.

During winter in Poznań (January–February), there is also high-pressure anticyclonic weather, which may explain the more significant number of the patients admitted to the emergency department with cardiovascular diseases. Concordance with the results of the Florentine and Poznań studies was demonstrated, in addition to the previously mentioned, and also with studies from Kaunas (Lithuania) between 1995–2000 [32], which documented an inversely proportional significant impact of the level of atmospheric air temperature on the incidence of myocardial infarction in the population aged 65–65 years. This was 84 (r = −0.048, *p* = 0.024), which was higher in winter than summer. On the other hand, the increase in the daily amplitude of atmospheric pressure caused a higher incidence of myocardial infarction in people of this age (r = 0.166, *p* = 0.001). At the same time, in Poznań, this adverse effect was visible only two days after the increase in the daily amplitude of atmospheric pressure. It was associated with an increased number of reports to the EDs.

In Poland, works on the impact of weather changes on the human body have a long tradition. In Poland, in 1976, the first post-war textbook on human bioclimatology for doctors was published by Jankowiak [33], characterising individual elements of the weather and the environment as potentially stimulating elements for the human body. Additionally, two publications published by Polish bioclimatologists in 1997 and 2007 [34,35] described the theoretical impact of individual climate and weather elements on health. The scientists also tried to present the methodology for calculating the effect of individual meteorological parameters on the load of the human body [35]. In 1968, two Polish scientists, Sładki and Żak [36] described the dependence of weather and climate changes on the transformation of connective tissue mucopolysaccharides. According to the authors, the changes in this tissue may be the genesis of malaise during a sudden change in weather. Skrobowski described the increase in blood pressure and the incidence of heart attacks in terms of meteorological parameters. He concluded that mechanical stimuli in the form of weather elements cause changes in the organs of the human body by the laws of physics [37,38]. Kuchcik showed the dependence of the number of deaths of Warsaw residents on the weather. She indicated that the Polish capital’s inhabitants’ health is affected by barometric systems such as the atmospheric fronts and air masses flowing over the territory of Poland, which result in days with very high or very low air temperatures, as well as in days with low atmospheric pressure [39,40]. She also determined that seasonal differences in the number of visits to the doctor’s clinic and the deaths in total due to cardiovascular diseases are much more minor compared to respiratory diseases. On the other hand, in the work of Lickiewicz, the occurrence of days with low atmospheric pressure values with the participation of the Foehn wind was described, in which the aggressive behaviour of patients intensified. In the case of air temperature (positive correlation) and humidity (negative correlation), weak but statistically significant relationships were found [41].

In the presented research about Poznań, attention was focused on the interaction between the day, season, and the average daily number of calls, which, thanks to the model used, could be given in actual and forecasted values. The average daily number of applications decreased on weekends, and the observed and predicted numbers were similar. Additionally, the marked decreases in the calls during the warm season and three days before an increase in the maximum daily air temperature during the warm season were successively identical, or almost the same for the forecast calls. The rate of change in the number of observed and predicted reports was also very similar in the case of an increase in the number of patients in the EDs during the winter, two days after an increase in the daily amplitude of atmospheric pressure, or a change in the daily air temperature on the date of diagnosis.

Other publications have also attempted to estimate the number of patients depending on the day, season, or meteorological parameters. Wargon et al. [42] showed that the day of the week has a powerful influence in modelling the number of patients reporting at EDs, but adding meteorological data to the model did not increase its sensitivity, which finally gave indications with an error of the number of patients reporting at the level of 4.2% to 14.4%. Additionally, the conclusions contained in the paper by Murtas et al. [43], regarding the number of visits to EDs depending on the day of the week, coincided with the previously described studies. The most frequent reports were on weekends for children and Mondays for adults and seniors. Festivities were associated with a decrease in ED visits between 13 and 28 (SE 1.45 and 1.98). In contrast, special celebrations such as New Year’s Day and 15 August were associated with the most significant decrease, which was of at least 42 ED visits. The results confirmed the role of the seasonal effects of the day of the week and the whole year, the relationship between the meteorological and environmental variables, and the number of ED visits. The results of these studies, carried out in five hospitals in Milan over the period of 2014–2019, did not show complete agreement with regard to the relationship between the meteorological parameters and the number of ED calls, but, for example, a high level of cumulative rainfall was associated with a significant decrease in visits to the emergency department in four hospitals, with a maximum decrease in visits of 0.31 per day for every 1 mm of rainfall. The authors defined a high demand for EDs as the days upon which the number of visits exceeded the median of the previous 31 days. The days were defined as green (level 1) if the number of visits exceeded the median by less than 5%, yellow (level 2) if this was between 5% and 10%, and red (level 3) if it was greater than or equal to 10%. An alert system in EDs has been proposed, using a predictive model based on previous patient visits.

Different conclusions were obtained from the research by Marcilio et al. [44]. In all the patient count models, introducing temperature data resulted in worse or similar predictability than that in the models with calendar variables alone. The prediction accuracy was better in the short-term (7 days earlier) than in the longer-term (30 days earlier). Similar conclusions were obtained in the Calegari study [45]. By analysing the performance of four prognostic models of predicting the number of ED patient reports in a hospital in Porto Alegre (Brazil), it was shown that, to model the number of patients with conditions defined as very urgent and urgent (the Manchester Triage System MTS scale), the SARIMA model is best used, in which in the ARIMA model (autoregressive integrated moving average), based on the phenomenon of autocorrelation, i.e., the dependence of the current variable value on its past values, the share of seasonality was taken into account in the forecast. Adding meteorological parameters to this model did not improve its performance. According to the authors, the use of a simple seasonal exponential smoothing (SS) model was the most advantageous for estimating the total number of patients (according to the Manchester Triage System MTS scale). In contrast, the work of the team of Jones et al. [46] indicated that, to better match the model to the sample forecasting the number of patients reporting to an ED, time series methods should be used. In the conclusions of his research, Armstrong [47] drew attention to the possibility of using the technique he described to model a wide range of multi-lag, non-linear relationships between air temperature and the number of deaths. 

Gasparini et al. [48] noted that most studies reported only heat-related effects and reported an increase in the excess mortality proportional to the extent of global warming in different climate change scenarios, or reported variations in both heat- and cold-related deaths, reporting the increase of mortality in the first case and its decrease when the temperature decreases. The authors also raised the difficulties of obtaining the results due to the variety of analytical projects. The advantages of the research conducted by Gasparini et al. were the number of almost 86 million analysed cases of deaths registered between 1984–2015 in 23 countries, and the possibility of a consistent comparison of data from hundreds of locations in different regions of the world, characterised by different climate, socio-economic, and demographic conditions, and different levels of development of their infrastructure and public health services. Gasparini et al. found that under high greenhouse gas (GHG) scenarios, most regions will experience a sharp increase in heat-related mortality, which will not be matched by the reduction in cold-related deaths, resulting in a significant positive increase in net mortality. 

Lo et al. [49] compared the performance of new models for monitoring heat-related deaths in the near-real-time regions of England, noting that over the years of 2011–2020, the estimates of heatwave mortality across the country were consistent with the excess deaths estimated by the British Health Security Agency. The method proposed by this team was superior in its ability to assess the after-effects of heat and control for other risk factors. It envisaged its use to estimate heat-related health burdens reliably in near-real-time and short-term projections.

The results of the Poznań study also confirmed the compliance of the number of actual EDs admissions with the forecasted ones, allowing for the use of the developed final model, enriched with weather parameters, for planning purposes. Still, it should be extended to include other variables that affect the population’s health. The study was treated as a pilot study, as the authors plan to expand the scope of the study and the follow-up period to 10 years. 

## 5. Conclusions

The results from this study indicate the influence of seasonality and meteorological parameters, and their changes on the number of registers for EDs in Poznań. The proposed final model allows for the short-term forecasting of the number of admissions to EDs and shows the consistency between the number of actual and predicted requests. However, although the final model was significantly better than the model containing only seasonal information (weekends and seasons), the changes in the number of patients were explained by the weather factors and seasonality in less than 17%. In addition to the estimated number of patients, depending on the seasonality and weather changes, models should also consider other factors: environmental, e.g., air quality, or socio-economic factors that determine the behaviour and health status of the population. Therefore, to increase the percentage of the explained variability in the number of patient reports, it is necessary to include additional factors that influence health status in the model.

For planning purposes, the nature of the hospital and its capacity, in terms of the number of beds and qualified medical staff, should also be taken into account. This is the first project in the history of Poznań to determine the impact of meteorological parameters on cases of cardiovascular diseases using a linear regression model. The findings extend knowledge on the effects of weather on health for the inhabitants of regions and cities in a moderate oceanic climate. Furthermore, they determine that the cold season, i.e., autumn and winter, is significant for reporting patients with cardiovascular diseases in this climate zone. It is also possible to determine the development of similar projects concerning Poznań.

## Figures and Tables

**Figure 1 ijerph-20-04838-f001:**
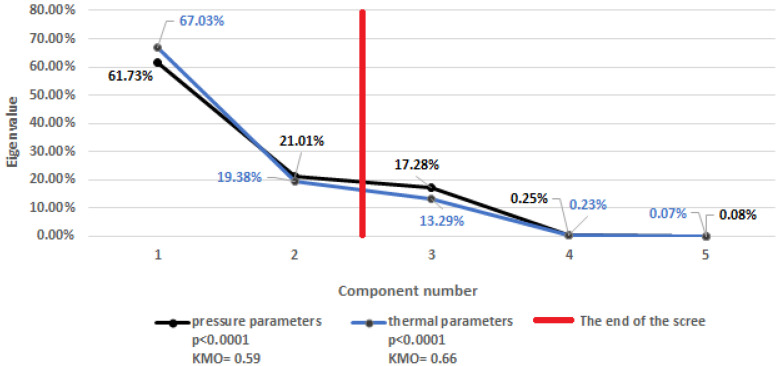
Scree plot showing the relationship between the number of components and the percentage of explained variance of meteorological parameters, together with the results of the Bartlett test and the value of the Kaiser–Mayer–Olkin coefficient.

**Figure 2 ijerph-20-04838-f002:**
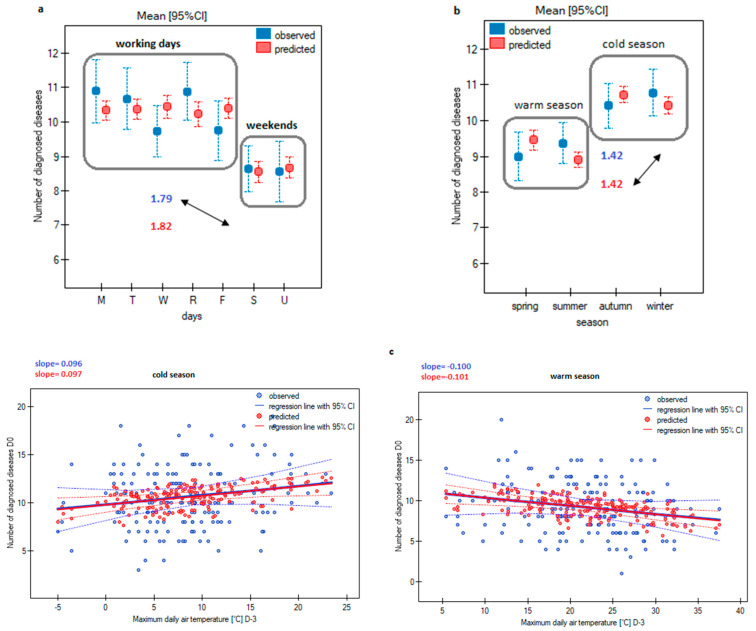
Summary of the number of diagnosed diseases and those predicted by the model, depending on: days of the week, season, maximum daily air temperature, interdiurnal change in air temperature, and the logarithm of the amplitude of daily atmospheric pressure. It was developed in the PQStat v1.8.4 program. (**a**) Averages determined depending on the day of the week with 95% confidence intervals for the observed number of calls and the number of calls predicted by the model. (**b**) Averages determined depending on the season with 95% confidence intervals for the observed daily number of calls and the daily number of calls predicted by the model. (**c**) Determined regression lines with a 95% confidence interval for the dependence of the daily number of calls on the maximum temperature recorded three days in advance in the summer and the cold season. (**d**) Determined regression lines with a 95% confidence interval for the dependence of the daily number of calls on the inter-day change in air temperature. (**e**) Persistent regression lines with a 95% confidence interval for the dependence of the daily number of reports on the logarithm of the daily amplitude of atmospheric pressure recorded with a two-day delay.

**Table 1 ijerph-20-04838-t001:** Meteorological parameters used in this study.

Thermal Parameters [°C]	Barometric Parameters [hPa] *	Hygrometric Parameter [%]
1. Average daily air temperature 2. Maximum daily air temperature 3. Minimum daily air temperature 4. Daily amplitude of air temperature 5. Interdiurnal air temperature changes	6. Average daily atmospheric air pressure 7. Maximum daily atmospheric pressure 8. Minimum daily atmospheric air pressure 9. Daily amplitude of atmospheric air pressure 10. Interdiurnal atmospheric air pressure changes	11. Relative air humidity

* atmospheric pressure data has been reduced to sea level pressure (SLP).

**Table 2 ijerph-20-04838-t002:** List of disease entities included in the analysed group of diseases of the cardiovascular system, along with their ICD-10 codes and the number of diagnose.

Disease Entity Code According to ICD-10	Type of Disease	Number of Diagnoses
I10	Essential hypertension	2254
I11	Hypertensive heart disease	24
I21	Acute myocardial infarction	477
I25	Chronic ischemic heart disease	26
I63	Cerebral infarction	441
I64	Stroke, not specified as haemorrhagic or infarct	384
Sum		3606
Average daily number of diagnoses	-	9.66 ≈ 10

**Table 3 ijerph-20-04838-t003:** Factor loadings for individual thermal and barometric variables in the first two principal components (PCs).

Factor Loadings	Principal Component 1	Principal Component 2
**Thermal parameters**		
Average daily air temperature	−0.985	0.081
Maximum daily air temperature	−0.997	0.055
Minimum daily air temperature	−0.916	0.130
Interdiurnal air temperature change	−0.224	−0.970
The logarithm value of the daily air temperature amplitude *	−0.707	−0.051
**Barometric parameters**		
Average daily atmospheric air pressure	−0.989	−0.067
Maximum daily atmospheric air pressure	−0.940	−0.288
Minimum daily atmospheric air pressure	−0.978	0.114
Interdiurnal air-pressure change	−0.433	−0.116
The logarithm value of the daily amplitude of atmospheric air pressure *	0.255	−0.964

* parameters with skewed distributions were normalised by logarithmic transformation.

**Table 4 ijerph-20-04838-t004:** Pearson’s linear correlation matrices for individual thermal and barometric variables. (**a**) Thermal variables. (**b**) Barometric variables.

**(a)**					
**Thermal Variables**	**Average Daily Air Temperature**	**Maximum Daily Air Temperature**	**Minimum Daily Air Temperature**	**Interdiurnal Air Temperature Change**	**Logarithm Value of the Daily Air Temperature Amplitude**
**Average daily air temperature**	**1.00**				
**Maximum daily air temperature**	**0.99**	**1.00**			
**Minimum daily air temperature**	**0.96**	**0.91**	**1.00**		
**Interdiurnal air temperature change**	**0.16**	**0.17**	**0.12**	**1.00**	
**Logarithm value of the daily air temperature amplitude**	**0.59**	**0.70**	**0.38**	**0.14**	**1.00**
(**b**)					
**Barometric Variables**	**Average Daily Atmospheric Air Pressure**	**Maximum Daily Atmospheric Air Pressure**	**Minimum Daily Atmospheric Air Pressure**	**Interdiurnal Atmospheric Air Pressure Change**	**Logarithm Value of the Daily Amplitude of Atmospheric Air Pressure**
**Average daily atmospheric air pressure**	**1.00**				
**Maximum daily atmospheric air pressure**	**0.97**	**1.00**			
**Minimum daily atmospheric air pressure**	**0.97**	**0.89**	**1.00**		
**Interdiurnal atmospheric air pressure change**	**0.33**	**0.29**	**0.29**	**1.00**	
**Logarithm value of the daily amplitude of atmospheric air pressure**	**−0.18**	**0.05**	**−0.38**	**−0.06**	**1.00**

**Table 5 ijerph-20-04838-t005:** Values of slope coefficients in seven regression models obtained by stepwise removal of variables.

*b-Coefficient*	D-3	D-2	D-1	D0	D + 1	D + 2	D + 3
**Intercept**	9.798	10.109	10.487	10.796	10.422	8.992	10.326
**Weekend**	x	x	−0.925	**−1.833 ***	x	1.266	x
**Season (cold season/warm season)**	1.512	1.315	1.554	**1.893 ***	1.566	1.474	0.557
**Maximum daily air temperature [°C]**	**0.098 ***	0.059	0.044	0.038	0.020	0.035	0.024
**Interdiurnal air temperature change [°C]**	x	0.167	x	**0.180 ***	x	x	x
**Average daily atmospheric air pressure SLP [hPa]**	x	x	x	x	x	x	x
**The logarithm value of the daily amplitude of atmospheric air pressure SLP [hPa]**	x	x	x	x	x	**0.504 ***	x
**Average daily relative air humidity [%]**	x	x	x	x	x	x	x
**Interaction: season * Maximum daily air temperature [°C]**	**−0.197 ***	−0.163	−0.164	−0.177	−0.150	−0.144	−0.102
**Interaction: season * Average daily relative air humidity [%]**	x	x	x	x	x	x	x

The values marked with * indicate the variant of the meteorological parameter used in the final model. The values marked with “x” indicate that the parameter is not used in the analysis.

**Table 6 ijerph-20-04838-t006:** Raw and standardised coefficients were obtained for the final regression model, together with an assessment of the significance of individual variables.

Variables	*b-Coefficient*	*p*-Value	Standardised b
**Intercept**	$b_{0}=9.296$	<0.0001	
$x_{1}=\;$ **weekend (no = 0/yes = 1)**	$b_{1}=-1.760$	<0.0001	−0.254
$x_{2}=\;$ **season (cold = 0/warm = 1)**	$b_{2}=1.867$	0.0221	0.298
$x_{3}=\;$ **maximum daily air temperature (°C) accelerated to 3 days (D-3)**	$b_{3}=0.095$	0.0138	0.274
$x_{4}=\;$ **interdiurnal air temperature change (°C) on the same day (D0)**	$b_{4}=0.143$	0.0186	0.115
$x_{5}=\;$ **the logarithm value of the daily amplitude of atmospheric pressure in two days lag (D + 2)**	$b_{5}=0.542$	0.0267	0.116
$x_{2}\cdot x_{3}=$ **interaction (product) of season parameters (0/1) and maximum daily air temperature (°C) with the acceleration of three days (D-3)**	$b_{6}=-0.198$	<0.0001	−0.748
**The rest of the model, including variables with a lag of one day (D + 1)**	$b_{7}=0.123$	0.0193	0.123

**Table 7 ijerph-20-04838-t007:** Comparison of the final model with the minimal model containing only seasonal information.

Determination Coefficient	F-Test
Final model: R^2^ = 0.165	*p*-value < 0.0001
Minimal model: R^2^ = 0.110	

## Data Availability

The data presented in this paper are available on request from the corresponding author. The data is not publicly available due to the concluded agreement on making statistical medical data available on particular terms—consent to share data National Health Fund/Narodowy Fundusz Zdrowia. Medical data from a database. Agreement of 5 August 2020, for the provision of statistical data—case register.

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
