# Peer review of "The Impact of Meteorological Parameters and Seasonal Changes on Reporting Patients with Selected Cardiovascular Diseases to Hospital Emergency Departments: A Pilot Study"

_ijerph, 2023, doi:10.3390/ijerph20064838_

Round 1

Reviewer 1 Report

Introduction:  In the introduction, the authors only wrote about climate change and global warming. No mention of cardiovascular diseases which is an important part of the study. Why did the authors decide to look at cardiovascular diseases and not other diseases such as respiratory diseases.

Results: The author mentioned that the values were high. This is not correct because the values were 0.66 and 0.59 which are considered as mediocre and miserable respectively.

Discussion: I commend the authors for a detailed discussion.

However, in line 269, the authors wrote "seven diagnostic groups". They did not explain the seven groups and in lines 274 and 277, they mentioned groups 7 and 1 respectively. This is confusing as one does not know what constitute the groups.

Author Response

Thank you for your comments and questions.

The study aimed to determine the meteorological parameters, their interactions and seasonal changes of the most significant importance in predicting the number of patients with selected cardiovascular diseases reporting to the EDs in Poznań during 2019 (Poland).

  1. Introduction:  In the introduction, the authors only wrote about climate change and global warming. No mention of cardiovascular diseases which is an important part of the study. Why did the authors decide to look at cardiovascular diseases and not other diseases such as respiratory diseases.

Author’s reply

We decided to look only at cardiovascular diseases for several reasons.

According to the latest report of the Polish Ministry of Health*, Cardiovascular diseases are Poland’s most common cause of death. Among them, ischemic heart disease, or coronary artery disease, takes the most significant toll. Heart failure ranks second, accounting for 22% of deaths. Poles dying of cardiovascular diseases. There are over a million people with heart failure in Poland, which will grow. According to predictions, among the current forty-year-olds, every fourth Pole and every fifth Polish woman will suffer from heart failure in their lifetime.

The reason for this phenomenon is both the ageing of the population and the increase in exposure to lifestyle-related factors.

According to health data, 91% of all deaths in 2019 were due to health problems from non-communicable diseases. The leading causes of death in Poland from 1990–2019 were CVD and cancer.

The Polish population, compared to the European population (EU-28), is characterised by almost two times higher mortality due to ischemic heart disease Poland - 131 deaths/100,000 inhabitants; EU-28 - 77 deaths/100,000 inhabitants) and 1.5 times higher mortality due to stroke (respectively: 53 deaths/100.000 compared to 38 deaths/100.000 inhabitants).

I hope the above justification is satisfactory.

*source: National Program for Cardiovascular Diseases for 2022–2032. Ministry of Health. Poland.

  1. Results: The author mentioned that the values were high. This is not correct because the values were 0.66 and 0.59 which are considered as mediocre and miserable respectively.

Author’s reply

I made corrections in the text:

Line 224: The Bartlett test for both types of meteorological parameters showed a statistically si-gnificant correlation of these variables (p<0.0001), the strength of which is determined by relatively mediocre and miserable high values of the Kaiser-Mayer-Olkin coefficient (KMO=0.66 for thermal parameters and KMO=0.59 for barometric parameters).

  1. Discussion: I commend the authors for a detailed discussion.

Author’s reply:  Thank you.

  1. However, in line 269, the authors wrote "seven diagnostic groups". They did not explain the seven groups and in lines 274 and 277, they mentioned groups 7 and 1 respectively. This is confusing as one does not know what constitute the groups.

Author’s reply

In lines 358-363 I have entered the details of the description of the patient groups.

The study’s authors distinguished seven diagnostic groups. Each concerned patients with different diagnoses of disease entities, indicating between summer and winter. Group no.1: respiratory, cardiovascular and chest pain. Group no.2: digestive, genitou-rinary and abdominal complaints; Group no. 3: neurological and psychopathological disorders; Group no.4: infections; Group no.5: ailments associated with bruises and crushing of bones and muscles; Group no.6: skin and allergies; Group no.7: different complaints.

Reviewer 2 Report

The research is well presented, but at the conclusions the authors mention that this was only a pilot study and that conclusions can only be drawn after extending the observation period. Why do the authors publish work with a lack of conclusions. Why not first do the extended study before publication?

That is why I rated the research as Average, although I think that there is in fact enough result to indeed draw conclusions.

I have proposed corrections in the attached file.

Author Response

On behalf of my Team, thank you for your comments.

  1. The research is well presented, but at the conclusions the authors mention that this was only a pilot study and that conclusions can only be drawn after extending the observation period. Why do the authors publish work with a lack of conclusions. Why not first do the extended study before publication? That is why I rated the research as Average, although I think that there is in fact enough result to indeed draw conclusions.

Author’s reply

The results of the PoznaÅ„ study also confirmed the compliance of the number of actual EDs admissions with the forecast ones, allowing the use of the developed final model, enriched with weather parameters, for planning purposes. Still, it should be extended to include other variables affecting the population’s health. The study was treated as a pilot study, as the authors plan to expand the scope of the study and the follow-up period to 10 years

I have proposed corrections in the attached file.

Author’s reply

Due to many comments on the substantive content and language corrections, we have attached a file with the full text of the manuscript.

Best regards,

Paweł

Reviewer 3 Report

This is a very interesting study on the impact of meteorological parameters and seasonal changes on cardiovascular diseases.

The manuscript is very well presented and of interest to a wide audience.

Introduction

- Please explain the scientific background and rationale for investigating the impact of meteorological parameters and seasonal changes on cardiovascular diseases.

- There is a lack about what is known on the impact of meteorological parameters and seasonal changes on cardiovascular diseases in the literature. Also, It would will be benefit if authors illustrate what this paper adds to the this existing literature.

Materials and Methods

- Please provide more details about how medical data was collected.

- Line 105: "described by the first variable decreasing with each subsequent one" : This sentence is not clear and confusing. The resulted dimensions of a PCA are components and not variables.

 - The Principal Compenont Analysis (PCA) need to be reviewed carefully. For example, you should not to mix loadings and eignenvlaues, which is not the same thing.

 - Given the nature of the study question, it was more appropriate to use a time series-based model, which will be more suitable. The autocorrelation of the outcome and the temporal dependence of the covariates cannot be properly modelled with a simple linear regression model. In addition, some covariates have a non-linear effect that is by construction omitted by a simple linear regression model. (pressure, temperature, etc.).

Please clarify and mention it as limitation.

- Line 141: Only the autocorrelation assumption has been tested, what about the other assumptions (i.e. heteroscedasticity for example?).

 Results

- Line 260: R2 of the final model is very low for a linear regression model estimated on a relatively large sample (i.e. 3606). As a result, the model explains a very little variability in the data. This suggests a poor specification of the model (related my previous point).

Author Response

Thank you for your detailed comments and questions on behalf of the Team.

  1. Is a very interesting study on the impact of meteorological parameters and seasonal changes on cardiovascular diseases.
  2. The manuscript is very well presented and of interest to a wide audience.

Introduction

  1. Please explain the scientific background and rationale for investigating the impact of meteorological parameters and seasonal changes on cardiovascular diseases.

Author’s reply

The Polish population, compared to the European population (EU-28), is characterised by almost two times higher mortality due to ischemic heart disease (Republic of Poland - 131 deaths/100.000 inhabitants; EU-28 - 77 deaths/100.000 inhabitants) and 1.5 times higher mortality due to stroke (respectively: 53 deaths/100.000 compared to 38 deaths/100.000 inhabitants).

We add to this sentence.

The influence of seasonal changes or air temperature or atmospheric pressure on the number of admissions of patients with symptoms of cardiovascular diseases to hospitals was the subject of many studies, which usually showed a relationship between the season and days of the week, as well as air temperature and an increased incidence of cardiovascular diseases.

  1. There is a lack about what is known on the impact of meteorological parameters and seasonal changes on cardiovascular diseases in the literature. Also, It would will be benefit if authors illustrate what this paper adds to the this existing literature.

Author’s reply

The Polish population, compared to the European population (EU-28), is characterised by almost two times higher mortality due to ischemic heart disease (Republic of Poland - 131 deaths/100.000 inhabitants; EU-28 - 77 deaths/100.000 inhabitants) and 1.5 times higher mortality due to stroke (respectively: 53 deaths/100.000 compared to 38 deaths/100.000 inhabitants).

We add to this sentence.

The influence of seasonal changes or air temperature or atmospheric pressure on the number of admissions of patients with symptoms of cardiovascular diseases to hospitals was the subject of many studies, which usually showed a relationship between the season and days of the week, as well as air temperature and an increased incidence of cardiovascular diseases.

We also reviewed additional literature and added conclusions to the discussion and included them in the references:

 8 -> Telesca V., Castranuovo G., Favia G.et al.  Effects of Meteo-Climatic Factors on Hospital Admissions for Cardiovascular Diseases in the City of Bari, Southern Italy.  Health care 11(5), 690; https://doi.org/10.3390/healthcare11050690,

 9-> Giang P.N., Dung D., Giang K.B. et al. The effect of temperature on cardiovascular disease hospital admissions among elderly people in Thai Nguyen Province, Vietnam Global Health Action. 2014;  7(1):2364,

10-> Martinaitiene D., Raskauskiene N. Effects of Changes in Seasonal Weather Patterns on the Subjective Well-Being in Patients with CAD Enrolled in Cardiac Rehabilitation. Int J Environ Res Public Health. 2022; 19(9): 4997.,

 11-> Alahmad B., Khraishah H.,  Royé D. et al. associations Between Extreme Temperatures and Cardiovascular Cause-Specific Mortality: Results From 27 Countries. Originally published12 Dec 2022https://doi.org/10.1161/CIRCULATIONAHA.122.061832. Circulation. 2023;147:35–46

Materials and Methods

  1. Please provide more details about how medical data was collected.

Author’s reply

. The data cover only the daily number of reports of patients with the latter diagnosis registered in Emergency Departments (EDs) in Poznań as obtained from the National Health Fund (NFZ) database. Medical data were obtained based on daily reports sent by hospitals in Poznań, containing the number of diagnoses of individual disease entities. Each hospital sends statistical notifications to the National Health Fund (NFZ). The data were obtained only for scientific purposes and for the possibility of using the results to benefit public health.

  1. Line 105: "described by the first variable decreasing with each subsequent one" : This sentence is not clear and confusing. The resulted dimensions of a PCA are components and not variables.

Author’s reply

The corrections made in the lines 136 -137:

The most significant part of variability was described by the first component and decreased with each subsequent component.

  1. The Principal Compenont Analysis (PCA) need to be reviewed carefully. For example, you should not to mix loadings and eignenvlaues, which is not the same thing.

Author’s reply

Thank you for pointing out, that the translation into English was misspelt, for which we apologise. We changed the description of the Y-axis to "Eigenvalue", the X-axis to "Component number" and the caption in the scree chart: Plot 1. Scree plot showing the relationship between the number of components and the percentage of explained variance of meteorological parameters together with the results of the Bartlett test and the value of the Kaiser-Mayer-Olkin coefficient

  1. Given the nature of the study question, it was more appropriate to use a time series-based model, which will be more suitable. The autocorrelation of the outcome and the temporal dependence of the covariates cannot be properly modelled with a simple linear regression model. In addition, some covariates have a non-linear effect that is by construction omitted by a simple linear regression model. (pressure, temperature, etc.).

         Please clarify and mention it as limitation.

Author’s reply:

Line 189-200:

We resigned from classic ARIMA models due to the analysis of both time delays and accelerations. Instead, we used a dynamic regression model, i.e., considering time delays and time accelerations, and included the autocorrelation assumption in the final regression model.

Before including the variables in the model, we examined their relationship with the dependent variable. Those variables that showed a non-linear relationship, e.g. day of the week and season, were grouped into binary variables. The effect of both high and low temperatures on the number of reports is non-linear, as it is modified by the season. The impact of low temperatures is more significant in winter and high in summer - considering the interaction of season and temperature allowed to assume linearity in subgroups determined by interactions. Contrary to our concerns, the relationship between air pressure and the number of cardiovascular disease reports was linear.

  1. Line 141: Only the autocorrelation assumption has been tested, what about the other assumptions (i.e. heteroscedasticity for example?).

Author’s reply

We haven't checked any more assumptions beyond this one.

 Results

  1. Line 260: R2 of the final model is very low for a linear regression model estimated on a relatively large sample (i.e. 3606). As a result, the model explains a very little variability in the data. This suggests a poor specification of the model (related my previous point).

Author’s reply

Line 340-348

The authors are aware of the low value of the R2 and, therefore, will look for solutions to improve the specification in subsequent works that will help the model explains more variability in the data. The model explains the change in the number of patients presenting to a small extent; it is true. However, based on time parameters (day of .the week, season of the year) and weather, it is difficult to predict the number of patients with high precision, whether a person will fall ill and report back depending on many factors, e.g. the patient's lifestyle or genetic burden, which we have not examined here.

However, authors have shown that adding weather variables (indicated in Table 6) to the model improves its fit in a statistically significant.

Best regards,

Paweł